# Conservation and Innovation: Versatile Roles for LRP4 in Nervous System Development

**DOI:** 10.3390/jdb9010009

**Published:** 2021-03-14

**Authors:** Alison T. DePew, Timothy J. Mosca

**Affiliations:** Department of Neuroscience, Thomas Jefferson University, Philadelphia, PA 19107, USA; alison.depew@students.jefferson.edu

**Keywords:** LRP4, Agrin, synaptogenesis, neuromuscular junction, brain, central nervous system, peripheral nervous system, *Drosophila*

## Abstract

As the nervous system develops, connections between neurons must form to enable efficient communication. This complex process of synaptic development requires the coordination of a series of intricate mechanisms between partner neurons to ensure pre- and postsynaptic differentiation. Many of these mechanisms employ transsynaptic signaling via essential secreted factors and cell surface receptors to promote each step of synaptic development. One such cell surface receptor, LRP4, has emerged as a synaptic organizer, playing a critical role in conveying extracellular signals to initiate diverse intracellular events during development. To date, LRP4 is largely known for its role in development of the mammalian neuromuscular junction, where it functions as a receptor for the synaptogenic signal Agrin to regulate synapse development. Recently however, LRP4 has emerged as a synapse organizer in the brain, where new functions for the protein continue to arise, adding further complexity to its already versatile roles. Additional findings indicate that LRP4 plays a role in disorders of the nervous system, including myasthenia gravis, amyotrophic lateral sclerosis, and Alzheimer’s disease, demonstrating the need for further study to understand disease etiology. This review will highlight our current knowledge of how LRP4 functions in the nervous system, focusing on the diverse developmental roles and different modes this essential cell surface protein uses to ensure the formation of robust synaptic connections.

## 1. Introduction

To perform the complex computations that underlie cognition, behavior, and emotion, neurons must communicate with each other in a reliable, but plastic, fashion. This information transfer is mediated by synapses—essential, specialized sites of intercellular communication in the nervous system. During development, these connections form between pre- and postsynaptic cells through a series of intricate and intertwined steps, ranging from recruitment of proteins that comprise neurotransmitter release sites and receptors, to transcriptional changes and synapse-to-nucleus signaling that engage long-lasting changes [1,2]. The success of each step is critical for the development of a functional connection. To ensure this, each aspect of synaptic development is carefully dictated by signaling molecules that coordinate development by triggering activation or deactivation of pathways. These signals transpire between synaptic partners to ensure precise synchrony across development. Oftentimes, a single protein regulates many downstream steps in this process, leading to the concept of a “synaptic organizer”. These nexus points of cellular signaling trigger events in multiple downstream pathways; as such, their perturbation leads to widespread defects in development due to the disruption of a core component in those multiple pathways. Such proteins use different mechanisms to initiate downstream developmental events but often rely on a common principle wherein extracellular signals between neurons initiate intracellular changes [3,4,5]. Consistent with this, cell surface receptors are well positioned to serve as these synaptic organizers: they can bind a ligand on the outside of the cell and then enact changes in multiple downstream signaling pathways within the neuron [6]. In the last decade, LRP4, a member of the low-density lipoprotein receptor family, has emerged as a synaptic organizer in the nervous system, functioning in a range of cellular processes to signal between cells during development. In the nervous system, LRP4 has been largely studied at peripheral synapses, where it is critical for synapse formation but more recently emerged as a synaptic organizer at central synapses, where its numerous roles remain incompletely understood. 

Following the first identification of LRP4 in 1998 [7], our understanding of its function has grown to encompass a wide range of cellular events, primarily during development. LRP4 is a single pass, type I transmembrane protein, and a member of the low-density lipoprotein-related receptor (LRP) family. It functions as a receptor for various ligands, including Wnt and Agrin [8]. Like other LRPs, LRP4 contains multiple LDLa repeats, as well as EGF-like and β-propeller domains, which are important for ligand binding to the extracellular domain of the protein (Figure 1). LRP4 also shares intracellular sequence motifs with other LRPs, including an NPxY motif, that are important for receptor endocytosis, trafficking, and intracellular signaling [9,10]. Biologically, LRP4 is involved in a myriad of processes, including limb development [11,12], craniofacial organogenesis [13,14], osteogenesis [15], kidney development [16], and neurodevelopment [17]. Loss of LRP4 in mice results in death due to paralysis, likely caused by a failure of the neuromuscular junction (NMJ) to develop [17], thereby preventing communication between motor neurons and muscle cells. This suggests that its role in synapse formation and organization is paramount for survival. Since its initial implication in nervous system development, work from a number of groups showed that LRP4 promotes neuromuscular junction formation and maintenance [17,18,19,20], peripheral nerve regeneration [21], central nervous system development [22,23,24], cognitive function and plasticity [25,26], and adult hippocampal neurogenesis [27] across different organisms and in different model systems. This indicates that these critical and diverse roles in neurodevelopment may be evolutionarily conserved. Further underscoring its importance in nervous system function and highlighting a potential link to human disease, the loss of LRP4 is linked to multiple neurodegenerative diseases including myasthenia gravis [28,29,30], amyotrophic lateral sclerosis (ALS) [31], and Alzheimer’s disease [32,33]. To begin to understand these roles, we will consider them in the context of three major categorizations, into which the functions for each of these roles can be placed: (1) whether LRP4 functions pre- or postsynaptically, (2) whether LRP4 functions in an Agrin-dependent or Agrin-independent manner, and (3) whether LRP4 functions in a cell-autonomous or non-cell autonomous fashion. In considering the diverse roles for LRP4, these categories (though neither essentially mutually exclusive of one another nor exhaustive) offer a useful conceptual framework to dissect our current knowledge of the different mechanisms of LRP4 function. Further, as new findings begin to elucidate the mechanisms of LRP4 function in central synapse development, exciting questions have emerged. How do these categories factor into LRP4 function at peripheral and central synapses? How are perturbations of LRP4 linked to human disease? While our understanding of these roles for LRP4 is ever growing, ongoing studies have provided tantalizing evidence for LRP4 as a new player in both development and disease at a variety of synapses.

## 2. LRP4 as a Critical Receptor for Agrin during NMJ Development

Arguably the most well-understood role, to date, of LRP4 is as a synaptogenic molecule at the mammalian NMJ. The NMJ is a highly specialized synapse where motor neurons contact muscle fibers. There, signals from the presynaptic motor neuron are transmitted to muscle via the neurotransmitter acetylcholine, which binds receptors on the postsynaptic muscle cell, triggering ionic influx, and resulting in contraction. The unique structure of the NMJ allows for fast, reliable neurotransmission to ensure each nerve impulse results in muscle contraction. Acetylcholine receptors (AChRs) are anchored on the postsynaptic muscle cell membrane in highly concentrated clusters which correspond to presynaptic sites of acetylcholine release [34,35]. This clustering represents a key event in synaptogenesis and synaptic development at the mammalian NMJ. During development, before the muscle cell becomes innervated by a motor neuron, AChRs are pre-patterned in an evenly distributed manner along the center of the muscle fiber. Arrival of the motor axon initiates cellular changes in the muscle via secreted motor-neuron derived signaling molecules that instruct clustering and anchoring of postsynaptic AChRs, as well as other aspects of postsynaptic differentiation, including transcriptional changes [36,37]. One such secreted signaling molecule is the proteoglycan Agrin, which is released from the motor neuron and can induce AChR clustering [38]. Agrin signals postsynaptic differentiation via activation of a postsynaptic receptor tyrosine kinase, MuSK (Muscle Specific Kinase) [39,40,41,42]. Though Agrin and MuSK were established as major synaptogenic molecules functioning in the same pathway, a direct interaction between them could not be shown [41], leading to the hypothesis that a third, yet-unidentified, molecule connected the two and serving as an Agrin receptor [41]. LRP4 became that likely candidate following the discovery in 2006 that it is expressed in muscles during development and its loss-of-function phenotypes mirror those of MuSK mutants: *lrp4* mutant muscles lack AChR clusters, and postsynaptic proteins including Rapsyn and Utrophin do not localize properly [17]. Furthermore, AChR clusters fail to form in *lrp4* mutant myotubes, either spontaneously or in response to Agrin stimulation [17]. Landmark work by the Mei and Burden labs subsequently showed that LRP4 can bind both Agrin and MuSK [18,19]. In this role, postsynaptic LRP4 in the muscle binds Agrin via its N-terminus, forming a tetrameric complex (two molecules of Agrin and two molecules of LRP4) and promoting the subsequent interaction between LRP4 and MuSK [43,44]. The ensuing downstream molecular cascade results in the dimerization and phosphorylation of MuSK [35,45] and subsequent phosphorylation of Dok7, which leads to transcriptional changes and clustering of postsynaptic proteins [46,47]. AChR clustering occurs through interaction with Rapsyn, a peripheral membrane protein which binds AChRs [48]. (Figure 2). Subsequent work identified additional regulators of LRP4, including Mesdc2 and CTGF, which promote LRP4 function through increased surface expression [49] and enhanced MuSK binding [50]. In this most well-understood role of LRP4, its function can be classified as postsynaptic, Agrin-dependent, and cell autonomous, according to the abovementioned categories. The discovery of this critical role of LRP4 established the missing link of mammalian NMJ formation and opened a new field of study to understand the downstream mechanisms of LRP4 function, the additional roles LRP4 engages in different processes at peripheral and central synapses, and the current categories by which we can begin to group those roles. 

## 3. Further Roles for LRP4 in the Peripheral Nervous System

The discovery of LRP4 as the receptor of Agrin to instruct postsynaptic differentiation via MuSK established the basis of our understanding of LRP4 function in the nervous system. However, importantly, it raised the tantalizing possibility that LRP4 could have additional roles at synapses. While postsynaptic NMJ differentiation remains its most studied role to date, further work identified other, earlier, mechanisms of LRP4 function at the NMJ (Figure 3). These roles contrast the well-known function of LRP4 discussed above, and further highlight the different categories of LRP4 function: throughout NMJ development and in contrast to its role in synaptic clustering as the Agrin receptor, LRP4 can also function at the presynapse, independently of Agrin, and in a non-cell autonomous manner. 

Early in development, before its role in postsynaptic differentiation, LRP4 is important for initial steps of synapse development. Prior to motor neuron arrival, LRP4 is required to instruct cellular events; *lrp4* mutant mice do not exhibit early cell-autonomous clustering of AChR receptors in the prospective synaptic region [17]. This indicates that LRP4 functions at the postsynapse to instruct pre-patterning of AChR in the muscle (Figure 3A). The mechanism of this earlier function of LRP4 differs from its later role in AChR clustering in that it is does not require Agrin binding but requires MuSK [51]. Thus, in the case of this pre-patterning role for LRP4, it functions at the postsynapse, independently of Agrin, and in a cell-autonomous manner. Beyond this postsynaptic role in the initial pre-patterning of AChRs, LRP4 plays additional roles in early NMJ development, but at the presynapse. In mice, loss of LRP4 results in excessive growth of motor axons along the muscle fiber, suggesting that LRP4 is required to signal the growth cone to stop and form adhesive connections [17,52,53] (Figure 3A). The absence of such a stop signal results in overgrowth of the motor neuron. While the exact mechanism of this role is unknown, it may be linked to the role of LRP4 in pre-patterning, as pre-patterned AChRs may serve as an indicator of the prospective synaptic region. In contrast to LRP4’s role in pre-patterning, however, this role as a stop signal is likely the result of a muscle to motor neuron signal. Specific loss of presynaptic LRP4 from the motor neuron does not produce an axonal overgrowth phenotype, suggesting that cell-autonomous LRP4 is not required for this context [35]. This indicates a more likely postsynaptic, Agrin-independent, and non-cell autonomous function. When juxtaposed, these roles highlight the intriguing complexity and importance of LRP4 function in utilizing different strategies to control temporally linked steps of neurodevelopment.

Once a connection is established, LRP4 continues to play a role in motor neuron development. In the absence of LRP4, motor neurons are unable to cluster active zone and vesicle proteins, indicating that LRP4 also serves as an early signal to induce presynaptic differentiation (Figure 3B) [17,52]. Intriguingly, though, these severe presynaptic effects are mediated by a non-cell autonomous mechanism of muscle-derived LRP4 [52], which requires that LRP4 enact these cellular events from across the synapse. Thus, LRP4 functions in *trans* as a retrograde signal to instruct presynaptic differentiation [54]. Furthermore, it functions independently of Agrin binding and MuSK activation, likely by interacting with a yet-unidentified receptor on motor axons [52]. Thus, while postsynaptic LRP4 is required, it does so by acting directly on the presynapse. This retrograde role for LRP4 is further supported by evidence that LRP4 is cleaved at the NMJ, its extracellular domain is sufficient to instruct NMJ formation, and it does not need to be anchored in the cell membrane to function [35,55,56,57]. The cleaved extracellular region of LRP4 thus functions as a ligand to instruct downstream recruitment of active zone and vesicle proteins [52]. The proteolytic mechanism by which LRP4 is cleaved in order to function as a retrograde signal is currently unknown, though it likely occurs via matrix metalloproteinase (MMP) activity [54]. Recent work at the *Xenopus* NMJ has identified a potential proteolytic mechanism. At this synapse, MT1-MMP is required for Agrin-LRP4 signaling, and cleavage of LRP4 by MT1-MMP is sufficient to produce the diffusible extracellular fragment of LRP4 to signal presynaptically [57]. Whether this mechanism extends to other species and at other synapses, however, remains unknown. In all, though these roles can be categorized into the three broad classifications as noted above; this particular example further highlights the complexity and versatility of LRP4 function. Furthermore, it presents a fascinating opportunity for future study to determine the mechanism by which retrograde LRP4 signals presynaptic differentiation. 

Beyond the non-cell autonomous function of muscle-derived LRP4 on the motor axon, LRP4 derived from the presynaptic motor neuron itself can also function cell autonomously. Though this pool of LRP4 is not essential for survival, it can promote development [54]. Genetic studies that removed LRP4 function specifically from one tissue, either muscles or motor neurons, allowed for the separation of these two pools and an assessment of how each contributes to development [54]. Interestingly, despite its importance as a postsynaptic molecule, loss of mouse *lrp4* from muscle alone results in viable pups with underdeveloped NMJs and immature AChR clusters that have formed but not achieved their fully developed adult form. However, loss of *lrp4* in both muscle and neuron results in perinatal death and NMJs resembling whole animal mutants, suggesting a role for neuronal LRP4 [54] or at least the combination of neuronal and muscle LRP4 in ensuring viability. Motor neuron-specific *lrp4* mutant mice are functionally similar to control mice, suggesting that neuronal LRP4 may play a role in development, but likely within the abovementioned combination or via a non-essential mechanism. Thus, non-muscle LRP4, likely from the motor neuron, may promote rudimentary AChR clustering in the absence of muscle LRP4, resulting in the immature AChR clusters observed in the muscle alone knockout [54]. This further suggests an intricate interplay between the two pools of LRP4 and highlights an interesting counterposition to the previous example: here, LRP4 would be presynaptically derived and function non-cell autonomously at the postsynapse initiate AChR clustering. How pre-patterning might occur in the absence of muscle LRP4 and prior to arrival of the motor neuron, though, remains unclear. Finally, despite working in *trans,* LRP4 function here is likely Agrin-dependent [54]. It is important to note that this may not serve as the only role for LRP4 in the motoneuron; a more thorough understanding of the phenotypic consequences of motoneuron *lrp4* loss will be important to determine if other, cell-autonomous, presynaptic functions exist. Regardless, these elements highlight a deep complexity to LRP4 with multiple, nuanced roles at the synapse.

While many of these functions of LRP4 require its activity as an Agrin receptor, a growing body of work in recent years highlighted the Agrin-independent roles of LRP4 function. Outside of the nervous system, LRP4 likely functions entirely independent of Agrin. Mechanistically, how might this occur? One possibility, drawing from developmental pathways, is that LRP4 functions in Wnt signaling pathways [15,58,59]. In bone and kidney, LRP4 works with Wnt ligands to promote development [13,15,16]. In these contexts, LRP4 is a Wnt signaling antagonist, either by binding Wnt directly, or by interacting with a Wnt inhibitor [15,58,59]. In the nervous system, Wnt proteins are essential for CNS synapse formation [60,61], AChR clustering at the vertebrate NMJ [62,63], and NMJ formation in *Drosophila* and *C. elegans* [64,65,66]. Consistent with these similarities, there is remarkable interplay between LRP4 and Wnts in the nervous system, particularly in AChR clustering in both mice and zebrafish [62,67,68]. At the mammalian NMJ, Wnts have complex and diverse roles in AChR clustering [67,69]. Two Wnts in particular, Wnt9a and Wnt11, induce AChR clustering, and this role is dependent on LRP4 and MuSK, but independent of Agrin [62]. This suggests that Agrin and Wnt function in distinct, parallel pathways to instruct clustering [62,68]. Whether LRP4 serves as a Wnt receptor here is unknown, however, though LRP4 can directly interact with Wnt9a and Wnt11 [62]. Furthermore, other Wnts can inhibit Agrin-induced AChR clustering, indicating crosstalk between Wnt and Agrin pathways, though it is unclear whether LRP4 is involved in this inhibition [67]. Due to these interesting intersections, the interplay of LRP4 and Wnt signaling, and indeed, the mechanistic basis for Agrin-dependent vs. Agrin-independent LRP4 function is an intriguing avenue for future investigations. 

Finally, LRP4 continues to function after synapse formation at the NMJ, with a critical role in synaptic maintenance. As whole-animal mutation of *lrp4* results in neonatal lethality [17], understanding its role in adulthood has not been trivial. Conditional knockout of LRP4 in muscle during adulthood, however, revealed critical roles in NMJ maintenance: loss of LRP4 in adulthood leads to fragmentation of AChR clusters, reduced neurotransmission, and decreased muscle strength [20]. This loss of LRP4 in the adult NMJ results in a reduction in synaptic Agrin levels, indicating that LRP4 may be required to stabilize Agrin [20]. Thus, LRP4 is essential to maintain the structure and function of the NMJ after development. These post-developmental roles for LRP4 stretch beyond NMJ maintenance as well. In zebrafish, LRP4 functions in the peripheral nervous system in response to injury by coordinating axon-glia interactions to ensure regrowth of follower axons [21] and does so independently of Agrin and MuSK, via an unknown mechanism [21]. Our understanding of these roles remains incomplete: they highlight the differences in Agrin-requirement in adulthood, but how these roles (or potential additional roles) function with respect to pre- versus postsynaptic or regarding cell autonomy, remains a topic for future investigation. Regardless, these roles indicate the importance of LRP4 to the nervous system throughout life, highlight the complexity in its myriad of roles, and raise the likelihood that it is a powerful substrate for future study for its therapeutic potential, as many neurological diseases arise during development [70].

## 4. LRP4: An Emerging Player in CNS Development

In recent years, a growing body of literature has expanded LRP4 function beyond the peripheral nervous system and into the central nervous system, both in development and adulthood. These studies have raised new questions about the downstream mechanisms of LRP4 and whether it uses similar mechanisms as peripheral synaptogenesis to instruct central synaptogenesis. Intriguingly, much like its peripheral roles, the same major categories (pre- vs. postsynaptic; Agrin-dependent vs. independent; cell autonomous vs. non-cell autonomous) continue to emerge as classifiers for LRP4 in the central nervous system in terms of their mechanistic functions. 

In the CNS, cell-autonomous developmental roles for LRP4 begin even before synapses form, specifically in hippocampal neurogenesis. In the adult hippocampus, neurogenesis increases by an unknown mechanism in response to an enriched environment; this may allow the brain to adapt to environmental changes [71]. New neurons integrate their synaptic connections into existing networks [72,73]. In mice, loss of LRP4 in neural stem/progenitor cells (NSPCs) prevents this neurogenesis after exposure to an enriched environment. In this role, LRP4 functions cell-autonomously in NSPCs in an Agrin-dependent manner [27]. The third category, that of pre- vs. postsynaptic, cannot be applied as these neurons have not yet formed connections. Unlike predominant examples from the NMJ, where LRP4 signals through MuSK, this pathway utilizes a different kinase, Ror2, which is closely related to MuSK [74] and likely functions in a similar manner. This unexpected role for LRP4 and Agrin in the brain raises interesting questions about how LRP4 functions outside of synapse development [27] while highlighting similar mechanistic options for LRP4 as in the periphery. 

The most studied roles of LRP4 in the CNS, however, mirror those most-studied in the periphery: the function of LRP4 in synaptogenesis. In the CNS, this was first uncovered in the *Drosophila* olfactory system [22]. The *Drosophila* LRP4 homologue shares 61% identity in its extracellular region with mammalian LRP4 [22], suggesting from a sequence similarity standpoint that the two could have similar functions in nervous system development. Indeed, in the *Drosophila* olfactory system, LRP4 is expressed presynaptically (Figure 4A) and instructs active zone formation in excitatory, but not inhibitory, neurons: loss of presynaptic *lrp4* results in fewer active zones and postsynaptic receptor clusters, as well as disruption of active zone ultrastructure. Loss of *lrp4* also leads to behavioral consequences—flies lacking LRP4 in olfactory neurons exhibit a complete loss of olfactory attraction behavior, corresponding to synaptic defects. Just as loss of *lrp4* decreases active zone number, overexpression leads to an increase in active zone number, demonstrating an instructive role for LRP4 in mediating synaptic contacts [22]. *Drosophila* lack clear homologues for both Agrin and MuSK [75], so these functions for *Drosophila* LRP4 must occur through a different mechanism than at the mammalian NMJ, highlighting an Agrin-independent aspect of this novel presynaptic function for LRP4. This raises the hypothesis that LRP4 may be the more evolutionarily ancient player in the Agrin-MuSK pathway during synapse development. The mechanism by which LRP4 functions in olfactory receptor neurons is incompletely understood, but it likely functions at least partly with the serine arginine protein kinase SRPK79D to ensure proper active zone number [22]. *lrp4* mutants exhibit an aberrant accumulation of the active zone protein Bruchpilot (Brp) in transverse nerves, which is reminiscent of the *srpk79D* mutant phenotypes [76,77]. It remains unknown whether this accumulation of Brp in the axon corresponds to the decrease in the number of synaptic active zones. Loss of *srpk79D* mirrors additional *lrp4* mutant phenotypes, and overexpression of SRPK79D is sufficient to rescue both synapse number and behavioral defects in the absence of LRP4. The downstream requirement for SRPK79D highlights a common theme of LRP4 function: many pathways function via a downstream kinase, whether MuSK, Ror2, or SRPK79D. In all, this role for *Drosophila* LRP4 in central synaptogenesis is particularly interesting for its contrast to LRP4 function at the mammalian NMJ; LRP4 functions presynaptically, cell autonomously, and independently of Agrin (and MuSK). Thus, despite divergent mechanisms, LRP4 function can still be categorized into these broad classifiers of function. Beyond this, *Drosophila* may provide further insights into additional, presynaptic and/or Agrin-independent roles for LRP4 at mammalian synapses. Whether LRP4 also has non-cell autonomous roles in *Drosophila,* like in mammals, remains unknown. Emerging work in *Drosophila* indicates another central synapse role for LRP4, in synaptic partner choice in the visual system. This further demonstrates its widespread importance in the developing brain and suggests that the roles of LRP4 in the olfactory system may also apply to other brain regions (though the nature of its cell autonomy or non-autonomy is currently unknown). Here, LRP4 is required for proper targeting of photoreceptor axons and may undergo regulation by another LDL family member, *lost and found*. Perturbation of *lrp4* results in defects in target selection, but whether LRP4 is also involved in other aspects of visual system development, including synapse formation and function, remains unknown [78]. As LRP4 is expressed widely throughout the *Drosophila* brain [22] and the mammalian literature suggests diverse function, understanding how LRP4 promotes *Drosophila* synaptic development function may provide further insights into its versatile roles. For instance, whether *Drosophila* LRP4 also functions at the NMJ or in glial cells to regulate synaptic biology or neuronal development is unknown and remains an active topic of investigation.

Beyond *Drosophila*, there is also a conserved requirement for LRP4 in regulating central synapse formation in the mammalian brain (Figure 4B). In cortical and hippocampal neurons, the loss of LRP4 results in decreased density of both excitatory and inhibitory synapses, along with fewer, but longer, primary dendrites [23,24]. Consistently, overexpression of LRP4 increases synaptic density and results in shorter, more numerous dendrites [23]. Here, LRP4 is enriched in both pre- and postsynaptic membranes, suggesting both pre- and postsynaptic roles in the brain [25], though it is not yet known which of these pools instructs which aspect or aspects of synapse development (and thus, how this role is classified). At mammalian central synapses (as at the NMJ), LRP4 functions in an Agrin-dependent manner: loss of *lrp4* blocks Agrin-induced increases in synapse number [24,79]. The involvement of MuSK, however, is less well understood. MuSK is required for proper dendritic branching in the brain, but not for normal synaptic density [24], suggesting that LRP4 could use both Agrin-dependent/MuSK-dependent and Agrin-dependent/MuSK-independent pathways. Interestingly, evidence from synaptic tracing experiments suggests that perturbation of *lrp4* may lead to both postsynaptic and presynaptic defects, as knockdown of *lrp4* in neurons resulted in fewer presynaptic partners [23]. Thus, like at the NMJ, LRP4 may function in multiple ways to instruct pre- and postsynaptic development, highlighting mechanistic similarity between peripheral and central systems as well as the cell-autonomy vs. non cell-autonomy comparison. The downstream mechanisms that propagate LRP4-involved signaling, however, are unclear. One speculation is that LRP4 regulates the cytoskeleton, as this could account for defects in both development of presynaptic terminals and dendrite morphogenesis. This is also supported by evidence that LRP4 overexpression decreases the mobility of dendrite terminals [23]. Insights from *Drosophila* may also inform further work in the mammalian brain; both systems exhibit similar synaptic phenotypes following loss of *lrp4*. Furthermore, serine arginine protein kinases, which likely interact with LRP4 in the fly brain, are conserved in mammals [22,80] and might also function with LRP4 in the mammalian brain. However, it is important to note that there are critical differences between LRP4 in the fly brain versus the mammalian brain in its Agrin-dependence and the synaptic specificity of the signaling (excitatory vs. inhibitory). A deeper understanding of the mechanisms that underlie both evolutionary realms of LRP4 function is an exciting area for future study and will likely yield fascinating insight into brain development. 

Synaptic roles in the brain for LRP4 may also persist into adulthood, much like in the periphery. LRP4 is expressed robustly in the hippocampus, olfactory bulb, cerebellum, and cerebral cortex [81,82]. In mice, loss of *lrp4* in all tissues except skeletal muscle results in defects in cognition and synaptic plasticity [25]. In the absence of LRP4, CA1 neurons of the hippocampus have decreased spine density, and exhibit corresponding defects in synaptic transmission and postsynaptic integration [25,26]. Currently, the underlying etiology of these defects is unknown. CNS LRP4 may further function during adulthood in synaptic maintenance. At the NMJ, such a role was determined using conditional removal of *lrp4* after the events of development and during adulthood to distinguish between developmental and maintenance roles [20]. The equivalent experiments in the CNS, however, have not yet been performed to assess cognition and plasticity and remain an area for future study. As global knockout of *lrp4* is lethal, conditional approaches have been used to study brain function in adulthood: by restoring LRP4 expression in muscle [25] of otherwise null *lrp4* mutants, adult mice that lack *lrp4* only in the CNS can be examined. However, these experiments have also not assessed whether CNS LRP4 is required for development or maintenance. Since neurodevelopmental disorders can arise when early events of synapse formation fail to occur, or occur incorrectly [83], a developmental defect remains a possible explanation. Recent work on glial LRP4, however, poses an alternative hypothesis. LRP4 is expressed in astrocytes and modulates glutamatergic transmission in hippocampal neurons by regulating ATP release. This effect is mediated by Agrin, though the requirement for MuSK is unknown [84]. In addition, astrocytic LRP4 also has a role in promoting dendrite arborization in neurons [85]. As such, disruption of glial LRP4 may manifest as defects in cognitive function and synaptic plasticity [25]. These interesting new findings on the role of astrocytic LRP4 present further exciting opportunities to study its glial roles, which likely have widespread importance throughout the brain. 

Taken together, the current state of the field highlights widespread and important roles for LRP4 in the central nervous system ranging from neurogenesis, to synapse formation, and even cognition. In many ways, these central roles of LRP4 mirror those of the NMJ, especially in the context of their classification into pre- vs. postsynaptic mechanisms, Agrin-dependent vs. Agrin-independent mechanisms, and cell autonomous vs. non-cell autonomous mechanisms. However, beyond this, LRP4 has important, emerging roles in glia as motivation for future study. While further investigation is needed to identify the precise and versatile mechanisms by which LRP4 instructs these processes, its importance in the brain is increasingly apparent. 

## 5. An Evolutionarily Conserved Role for LRP4 in the Nervous System? 

How synapses evolved from simple groupings of proteins in early prokaryotes to the elegant, complex structures that underlie the complexity of the human brain remains a tremendous unanswered question. Synaptic proteins exhibit a remarkable degree of conservation across eukaryote species [86,87] and a number of mechanisms that instruct synapse development are conserved [88,89] in evolutionarily distant systems. Studies of LRP4 across species and systems present an opportunity to consider the evolutionary past of LRP4, which can provide insight into its roles in the nervous system. First, how did LRP4 evolve as a critical element of neurodevelopment? The LDL-receptor genes are an evolutionarily ancient family that appear with the first multicellular organisms [90]. LRP4 is likely a more recent member of this family. Organisms including *C. elegan*s have LRP homologues that more closely resemble LRP1 and LRP2, but no explicit homologue of LRP4. Protein BLAST reveals homologues of LRP4 in more complex invertebrates, including mollusks (such as *Aplysia californica* and *Octopus bimaculoides*), arthropods, and echinoderms (which all largely resemble *Drosophila* LRP4) [75]. The *Drosophila* homologue in turn, shares sequence homology with the extracellular portion of mammalian LRP4, but lacks the same intracellular motifs [8,22] (Figure 5). Despite these differences, LRP4 functions in both vertebrate and invertebrate nervous systems at central and peripheral synapses. Across these species, LRP4 signals aspects of synapse development, albeit in various ways. This indicates the evolutionary importance of the receptor, but how did its roles change during evolution? In *Drosophila*, the LRP4 protein localizes predominantly presynaptically, in contrast to mammals where it is predominantly postsynaptic [81]. This demonstrates how one classifier of LRP4 function, its expression at the synapse, emerged over the course of evolution. Further, in mammals, LRP4 functions in both peripheral and central synaptogenesis via an Agrin-dependent mechanism. Interestingly, the *Drosophila* genome does not contain obvious homologues to either Agrin or MuSK, suggesting that the Agrin-dependent functions of LRP4 also arose with evolution of a more complex nervous system. Yet, LRP4 shares a similar mechanism in flies in that it signals through a downstream kinase. This likely indicates that the mechanism of LRP4 function shifted over time but retained certain tenets of function. These considerations also raise interesting questions of whether mammalian LRP4 has retained some of its more ancient roles, including presynaptic roles in development, and perhaps signaling via a serine arginine protein kinase. 

**Figure 5 jdb-09-00009-f005:**
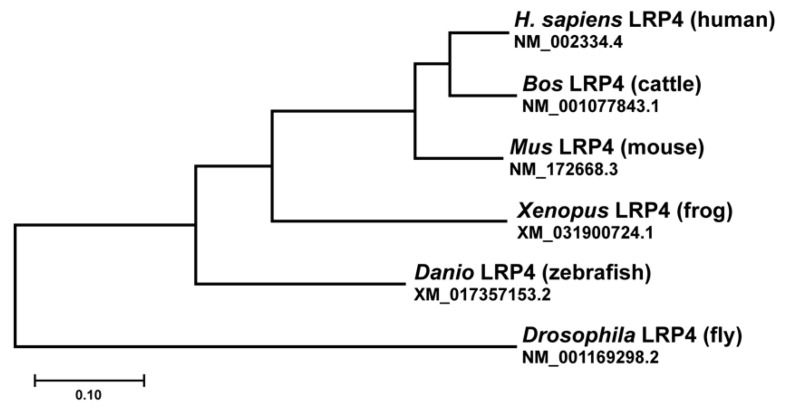
Phylogenetic tree of LRP4. LRP4 is conserved across species, ranging from invertebrates to humans. Despite its functional similarities, *Drosophila* LRP4 shares the least evolutionary history with other LRP4s. It shares 61% identity of the extracellular domain with human LRP4 but differs in the intracellular region. Evolutionary history was inferred by using the Maximum Likelihood method and Tamura-Nei model. Multiple sequence alignment was conducted using MUSCLE. Alignments and evolutionary analyses were conducted in MEGA X. Branch lengths are measured in the number of substitutions per site [91,92,93].

Additional roles for LRP4 outside the nervous system may provide further insight into its evolutionarily conservation. LRP4 has conserved roles in regulating Wnt signaling throughout the body [15,58,59], which have also been shown in the nervous system and further highlight Agrin-independent functions. Wnts are essential for synapse formation, brain development, and organization [60,61,94]. These Wnt dependent pathways may be more ancient, as they have been noted in invertebrates [95], while Agrin-dependent synaptogenic pathways have not [96]. This may also suggest that the roles of LRP4 outside the nervous system, where it functions with Wnt signaling, are older than the Agrin/MuSK-dependent pathway. Work on zebrafish NMJ development highlighted divergent mechanisms for Wnt, LRP4, and MuSK pathways [68]. Like the mammalian NMJ, the zebrafish NMJ exhibits pre-patterning of AChRs, but zebrafish LRP4 is not required for AChR pre-patterning. However, zebrafish do require LRP4, Agrin, and MuSK for synapse formation [68]. MuSK shares a similar cysteine rich domain with the Wnt receptor Frizzled [74,97], suggesting that MuSK may also serve as a Wnt receptor, potentially in this role. The conservation of some, but not all, of these pathways raises intriguing questions about the evolutionary history of LRP4. As LRP4 is expressed in central nervous system synapses and has been implicated in CNS synapse formation, it remains an interesting and open question whether CNS Wnt-dependent synapse formation requires LRP4, perhaps similar to the role of LRP4 in Wnt-induced AChR clustering at the NMJ [62]. 

## 6. From Development to Disease

Underscoring the importance of synaptic connections to all elements of neuronal function, the loss or mutation of genes that encode synaptic proteins are implicated in a range of neurodevelopmental, neurological, and neurodegenerative disorders [98,99]. These disorders can aggregately be classified as synaptopathies, where genetic mutations in or affecting synaptic proteins cause severe neurological and developmental disorders. Often, disruption of a single aspect of the complex synaptic machinery is sufficient to cause widespread changes in the brain, leading to disease [83]. Given the essential role for LRP4 in proper nervous system development and mature function, it is reasonable to hypothesize that the loss or perturbation of LRP4 may underlie some of these synaptopathies. Consistent with this hypothesis and given its well-known roles in the peripheral nervous system and NMJ function, recent work implicated LRP4 in disorders of the motor unit, specifically myasthenia gravis (MG), congenital myasthenic syndrome (CMS), and amyotrophic lateral sclerosis (ALS). 

Motor disorders often stem from mechanisms that result in impaired NMJ function or the breakdown of essential components that contribute to the normal function of the NMJ. Myasthenia gravis (MG) is an immune disorder where the body produces antibodies against proteins required for NMJ function, resulting in muscle weakness [100]. Often, these antibodies are against the AChR or MuSK, and loss of these proteins results in fragmentation of the NMJ [100]. More recent work implicated LRP4 as an additional player in MG pathology; MG patients that displayed symptoms consistent with the motor disorder, but were double seronegative against MuSK and AChR, instead had autoantibodies against LRP4 [28,101]. Experimentally, mice immunized with anti-LRP4 antibodies exhibited MG-associated symptoms including muscle weakness, as well as fragmented NMJs [29], suggesting a causative relationship between the loss of LRP4 and MG. Analogous symptoms also occur in congenital myasthenic syndromes (CMS), which are caused by mutations in genes important for NMJ function or maintenance [102]. LRP4 is implicated as a CMS disease gene, but unlike MG, this is likely due to direct mutations in the *lrp4* gene (and not auto-immune targeting of the LRP4 protein). In CMS patient populations, two *lrp4* mutations were identified which disrupt binding of LRP4 to Agrin and MuSK [103]. Thus, LRP4 is implicated in two similar motor disorders, though via different mechanisms, both resulting in reduced NMJ function. It remains unclear, though, how the different categories of LRP4 mechanistic function interact with their potential roles in motor disorders like MG. This will remain an interesting mechanistic question for future study. Mutation of LRP4 in humans, though, does not always affect synapse development. Patients with Cenani-Lenz syndrome, an autosomal recessive disorder which results in syndactyly and oligodactyly and is caused by homozygous, mutation of LRP4, have seemingly normal NMJs, despite loss of LRP4 function [16]. One explanation for this is compensation by LRP10, which is similar to LRP4 and may have resulted from a recent gene duplication [104]. Given these roles for LRP4 in disorders of the NMJ, ongoing work continues to target the Agrin-LRP4-MuSK pathway as a potential therapeutic target [105]. 

An auto-immune mechanism with LRP4 may also function in the motor disorder amyotrophic lateral sclerosis (ALS), as some ALS patient populations also have LRP4 autoantibodies [31,106]. ALS is a neurodegenerative disorder characterized by progressive muscle weakness leading to eventual paralysis and death [107]. Some ALS patient populations are seropositive for LRP4 autoantibodies in both serum and CSF samples, but their symptoms do not overlap with MG [31,106]. Though implicated, it remains unclear if these antibodies contribute to disease, and an active question is whether the perturbation of LRP4 via autoantibodies leads to the denervation of muscle observed in ALS. Moreover, how do autoantibodies produced in MG and ALS differ, and how do they differentially regulate LRP4? Given the role for LRP4 in the CNS, it is further possible that autoantibodies against LRP4 may contribute to upper motor neuron dysfunction, though not understood [106]. Future studies are needed to test the implication of LRP4 in ALS pathogenesis as it may have a potential role as a therapeutic target.

Finally, emerging roles in CNS function have highlighted a potential role for LRP4 in Alzheimer’s disease (AD). AD is characterized by lesions in the medial temporal lobe and cortical areas of the brain, as well as degeneration of neurons and synapses [108], and is one of the most common causes of dementia. Substantial work on the mechanisms of AD supports a hypothesis where the accumulation of amyloid-β (Aβ) in the brain contributes substantially to disease pathology. Aβ is produced by proteolytic cleavage of amyloid precursor protein (APP). Mutations that interfere with this cause the production of toxic Aβ that can form pathogenic plaques [108]. APP is regulated in part by ApoE (Apolipoprotein E), which is in turn regulated by members of the LDLR family [109]. It is this link to APP that may serve to connect LRP4 in its synaptic role with AD. In motor neurons, LRP4 can bind ApoE to promote neuronal cell viability [110]. During NMJ development, LRP4 and Agrin both interact with APP to promote AChR clustering. It is currently unknown whether this mechanism may also be present in the CNS [32]. As secreted APP functions as a ligand for GABA_B_ receptors in regulating synaptic transmission [111] and APP can function as a Wnt receptor [112], this raises the possibility of a relationship between synaptic proteins, LRP4, and AD. The mechanistic link, however, remains unknown, as does how Alzheimer’s progression fits into the established LRP4 functional categories (though a link to Agrin is strongly possible). A further role is supported by a function for astrocytic LRP4 in Aβ clearance. Members of the LDLR family including LDLR and LRP1 are involved in astrocytic clearance of amyloid-β [104,113]. LRP4 also functions in amyloid-β clearance [33] and is expressed at higher levels in astrocytes than other members of the LDLR family. In this role, LRP4 promotes Aβ uptake by astrocytes likely by serving as a receptor for ApoE (as dot blot assays demonstrate that purified LRP4 ectodomain and human ApoE can physically interact), and therefore may play an important role in response to AD progression. In mouse models of AD, loss of LRP4 in astrocytes exacerbated disease pathogenesis [33]. Furthermore, postmortem brain tissue from AD patients exhibited decreased levels of LRP4 [33]. This body of evidence indicates the potential for LRP4 as a therapeutic target, and the need for further study of the roles of LRP4 in Alzheimer’s disease, and the CNS. 

## 7. Conclusions

Once known in the nervous system only as the Agrin receptor at the NMJ, LRP4 has become a major player in various developmental processes and as an emerging neurodegenerative disease risk gene. As ongoing work demonstrates novel roles for this receptor in the CNS, its importance has only grown. In its various roles, LRP4 is defined by several categories of action: functioning at the pre- or postsynapse, with or without Agrin, and in a cell autonomous or non-cell autonomous manner. The emerging role of LRP4 as a synaptic organizer in the brain presents fascinating opportunities for future study, and new series of exciting questions about the nervous system: How does LRP4 instruct synapse formation in the brain? Does it function via conserved mechanisms in peripheral and central synaptogenesis? Does LRP4 have therapeutic potential in neurodegenerative disorders? These latest discoveries present new opportunities to understand synapse formation in the brain, the role of LRP4 in disease, and the conservation of developmental mechanisms. Further work on LRP4 is necessary to address these questions in our pursuit towards understanding the nervous system. However, indeed, as the mechanisms for LRP4 function in the CNS are discovered, future studies will investigate its potential as a therapeutic target in a multitude of neurological disorders, seeking to connect synaptic function with disease progression.

## Figures and Tables

**Figure 1 jdb-09-00009-f001:**
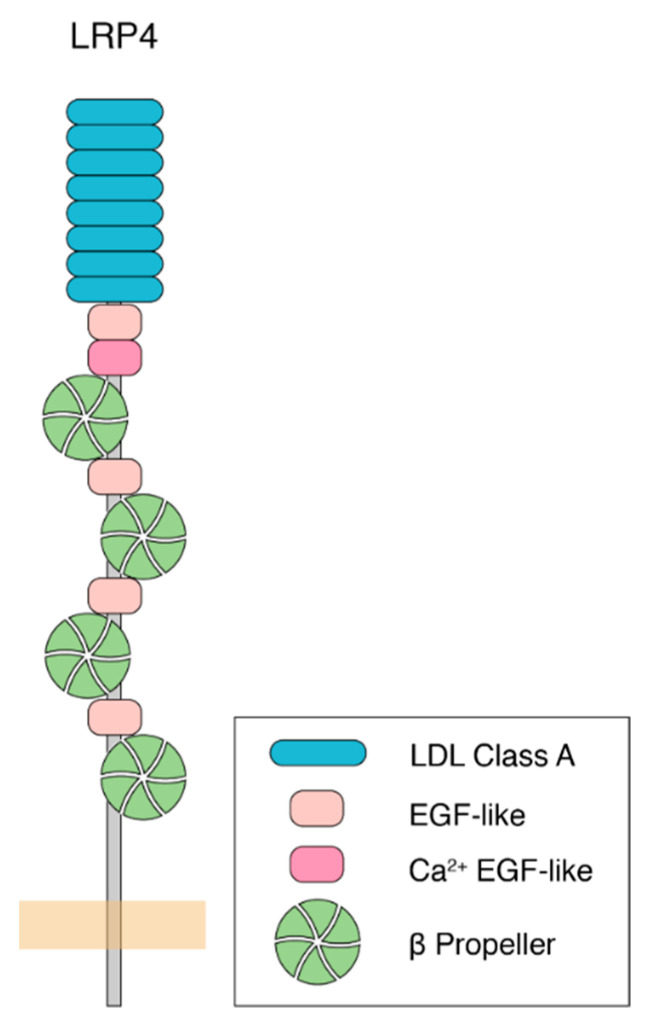
Molecular structure of human LRP4. LRP4 is a single pass, type I transmembrane protein. The large extracellular region, capable of ligand-binding to Agrin, Wnt, and other proteins, contains LDL repeats (blue), EGF-like domains (pink), and β-propeller domains (green). The intracellular portion is a comparatively shorter tail that contains an NPxY motif and a PDZ-binding motif, which are important for protein interactions and receptor endocytosis. While most of the extracellular portion is shared among species, other mammalian forms have an additional EGF-like domain after the final β-propeller domain. On the intracellular side, mammalian forms of LRP4 share the NPxY and PDZ-binding motifs, though these are lacking in the *Drosophila* homologue.

**Figure 2 jdb-09-00009-f002:**
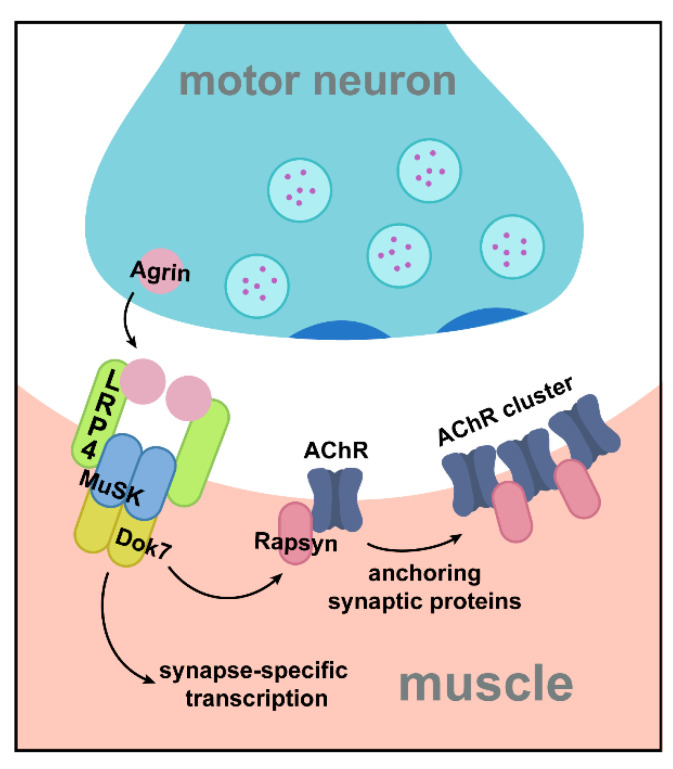
LRP4 is essential for postsynaptic differentiation at the mammalian NMJ. In its most well-known role at the mammalian NMJ, LRP4 serves as the receptor of Agrin. Agrin is a secreted proteoglycan, derived from the motor neuron, that traverses the synaptic cleft before binding to postsynaptic LRP4. Binding of Agrin to LRP4 induces the formation of a tetrameric complex (containing two molecules of Agrin and two molecules of LRP4) that promotes association with and activation of the receptor tyrosine kinase MuSK. Dimerization of MuSK subsequently signals downstream through Dok7 to instruct aspects of postsynaptic differentiation, including the anchoring of proteins like AChR via interaction with Rapsyn, and transcription of genes encoding synaptic proteins at subsynaptic nuclei.

**Figure 3 jdb-09-00009-f003:**
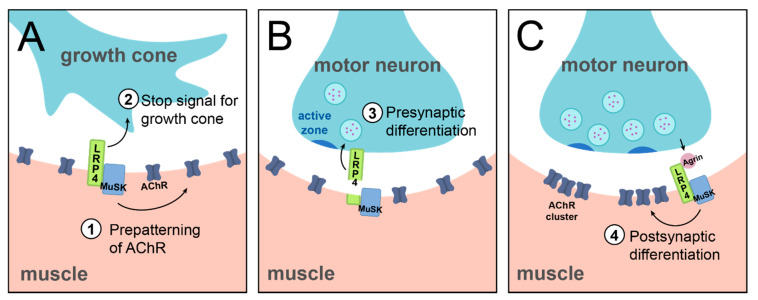
Muscle LRP4 regulates multiple steps in mammalian NMJ development. At the mammalian NMJ, LRP4 instructs multiple steps of development both in *cis* and in *trans* to regulate pre- and postsynaptic differentiation. Prior to the arrival of the motor axon (**A**), LRP4 is required for pre-patterning of acetylcholine receptors in the prospective synaptic region (role 1). Upon arrival of the motor axon, LRP4 is required as a stop signal for the growth cone to arrest further travel along the muscle fiber (role 2). This role is important for the initiation of adhesive contact between the motoneuron and the muscle. Once the growth cone has transitioned to a growing synaptic contact, LRP4 then instructs presynaptic differentiation (**B**, role 3), likely by the cleaved extracellular domain of LRP4 binding a yet-unknown receptor on motor axons to signal downstream clustering of vesicle and active zone proteins. LRP4 also signals postsynaptic differentiation (**C**, role 4) by binding Agrin and stimulating MuSK activation (discussed above; see Figure 2). For simplicity, this diagram represents a dimerized LRP4-agrin interaction (as shown in Figure 2) and represented in [44].

**Figure 4 jdb-09-00009-f004:**
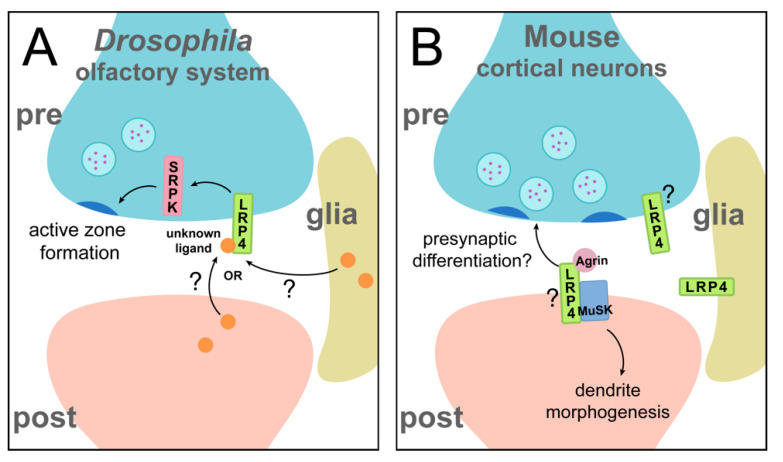
Models of LRP4 involvement in central synapse development. LRP4 is required for synapse development in both invertebrate and vertebrate central nervous systems. In *Drosophila* (**A**), which lack both Agrin and MuSK homologues, LRP4 functions presynaptically in the olfactory system to instruct active zone formation. The ligand of LRP4 (and its exact source) remain unknown in this model. Two potential sources, a postsynaptic partner or glia, are indicated in this diagram. To instruct presynaptic changes, LRP4 functions through the downstream kinase, SRPK79D. In mouse cortical and hippocampal neurons (**B**), LRP4 functions in a similar manner to regulate synaptic density. It is unknown whether LRP4 functions primarily at the pre- or postsynapse; for the purposes of this model, it is shown as postsynaptic but could also function presynaptically. In these contexts, LRP4 induces presynaptic differentiation via an Agrin-dependent mechanism and regulates dendrite morphogenesis through a MuSK-dependent mechanism. In addition, LRP4 has important glial roles (in regulating growth and neurotransmission) and may be functioning in this role or in currently undiscovered functions.

## Data Availability

No new data were created or analyzed in this study. Data sharing is not applicable to this article.

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
