# Peer review of "Conservation and Innovation: Versatile Roles for LRP4 in Nervous System Development"

_jdb, 2021, doi:10.3390/jdb9010009_

Round 1

Reviewer 1 Report

The manuscript by DePew and Mosca is a very comprehensive and well written review on the roles of Lrp4 in the nervous system and its role in different diseases. I do not have any further suggestions for this manuscript.

Reviewer 2 Report

Conservation and Innovation: Versatile Roles for LRP4 in Nervous System Development

Alison T. DePew and Timothy J. Mosca

This is a nice and comprehensive review manuscript by A. T. DePew and T. J. Mosca that focuses on the diverse roles of LRP4 during synaptic formation and development. The topic is of high interest for the scientific community and the manuscript represents a significant contribution to the field as it covers functions of LRP4 with great detail, which are usually missing in other manuscripts that review LRP4 function. This review manuscript is balanced and comprehensive, providing solid new information and identifying knowledge gaps.

Specific comments:

For role 1, the authors describe on Page 5, line 180: “this pre-pattering role for LRP4, it functions at the postsynapse, independently of Agrin, and in a cell-autonomous manner.” However, a few paragraphs later on page 6, line 224 they mention: “Loss of mouse lrp4 from muscle alone results in viable pups with underdeveloped NMJs and immature AChR clusters”, in addition to “non-muscle LRP4, likely from the motor neuron, may promote rudimentary AChR clusters in the absence of muscle LRP4” (line 230), which appears somehow unexpectedly and makes role 1 more complicated. Authors provide an interesting discussion, but it seems misplaced and disconnected. The reader would benefit if the authors could connect these two sections in a better way to prevent any unexpected confusion.

Page 5, line 189: “pre-patterning, this signals from” unclear what “this” means

Page 5, line 190: the authors describe role 2 as “postsynaptic, Agrin-independent, and non-cell autonomous function.” However, from the information provided in that section, it is not clear why a presynaptic, cell-autonomous role of LRP4 can be ruled out.

On page 7, section 4 describes the role of LRP4 in neurogenesis in mammals, it then describes synapse development in Drosophila, and it concludes describing synapse development in mammals. The authors may consider moving the neurogenesis section towards the end, so all sections describing roles of mammalian LRP4 are better connected.

At the end of page 8, the authors describe a new role of LRP4 in the Drosophila visual system. It would be interesting to mention if any of the phenotypes describe before at the NMJ or in the olfactory system are also observed in the visual network.

For section 4, it would be interesting to discuss whether the role of LRP4 in neurogenesis and in glial cells may be conserved in Drosophila. It is also not mentioned whether LRP4 has any function at the Drosophila NMJ.

For figure 4 on page 9, it is not addressed why a glia cell is shown as a possible player for Drosophila synapses but not shown for mammalian ones.

On page 11, lines 441-442: “In Drosophila, LRP4 is expressed predominantly presynaptically, in contrast to mammals where it is predominantly postsynaptic.” The authors should make clear whether the amount of mRNA/protein is meant, or alternatively, penetrant levels/impact of the described phenotypes. A corresponding citation here seems to be missing.

Section 5 on pages 10-11 includes an interesting discussion about LRP4 evolution among different species. It would be beneficial to include the most ancient organism with known LRP4 to prevent the impression that Drosophila is the first organism with the most ancient LRP4.

In the last paragraph of section 6, the authors include an interesting discussion about LRP4 and Alzheimer’s disease. The reader would benefit from more clarification about interactions between LRP4 and ApoE, as some of them occur in neurons and others in glia cells, and it is unclear what has been experimentally tested and what is hypothesized (for example in line 562: “likely by interacting with ApoE”).

The use of the three major categorizations (pre vs. postsynaptic, agrin dependency, and cell vs. non-cell autonomous) is an interesting way to understand LRP4 functions. The authors may consider the possibility of including a table that summarizes the different roles and crucial findings that led to the corresponding categorization.

Reviewer 3 Report

This is a highly interesting, well-written and complete review on the intriguing role of LRP4 in synaptic development and maintenance. I very much enjoyed reading it.

The only major comment is that the review is perhaps a bit longish, and could be shortened here and there.

Further (minor) points:

- line 64: abbreviate NMJ on first mentioning. Check manuscript for all abbreviations.

- page 4 top and Fig 2: perhaps provide more detail here in that agrin-LRP4 form a tetrameric conformation complex that induces dimerization of MuSK. In other words, mention/illustrate the necessary dimerizations.   

- line 402-404: This sentence is unclear/wrong

- Fig 5: please insert the common names (e.g. zebrafish, cow etc.) Bos and Danio will not be immediately clear to non-specialists.

- line 503: ‘additional mayor player’ I would leave out ‘major’ because LRP4 antibody positive MG is still very rare and not undebated in the field
